# Effect of a Multiple Reduction Die on the Residual Stress of Drawn Materials

**DOI:** 10.3390/ma14061358

**Published:** 2021-03-11

**Authors:** Jeong-Hun Kim, Chang-Hyun Baek, Sang-Kon Lee, Jong-Hun Kang, Joon-Hong Park, Dae-Cheol Ko

**Affiliations:** 1Precision Manufacturing Systems Division, Pusan National University, 2, Busandaehak-ro 63beon-gil, Geumjeong-gu, Busan 46241, Korea; kimjh83512@pusan.ac.kr (J.-H.K.); baek1201@pusan.ac.kr (C.-H.B.); 2Ultimate Manufacturing Technology Group, Korea Institute of Industrial Technology, 320, Techno Sunhwan-ro, Yuga-myeon, Dalseong-gun, Daegu 42990, Korea; sklee@kitech.re.kr; 3Department of Aeromechanical Engineering, Jungwon University, 85, Munmu-ro, Goesan-eup, Goesan-gun, Chungbuk 28024, Korea; jhkang@jwu.ac.kr; 4Department of Mechanical Engineering, Dong-A University, 37, Nakdong-daero 550beon-gil, Saha-gu, Busan 49315, Korea; acttom@dau.ac.kr; 5Department of Nanomechatronics Engineering, Pusan National University, 2, Busandaehak-ro 63beon-gil, Geumjeong-gu, Busan 46241, Korea

**Keywords:** drawing process, multiple reduction die (MRD), residual stress, X-ray diffraction (XRD), deep neural network (DNN)

## Abstract

Residual stress may influence the mechanical behavior and durability of drawn materials. Thus, this study develops a multiple reduction die (MRD) that can reduce residual stress during the drawing process. The MRD set consists of several die tips, die cases, and lubricating equipment. All the die tips of the MRD were disposed of simultaneously. Finite element analysis of the drawing process was performed according to the reduction ratio of each die tip, and the variables in drawing process with the MRD were optimized using a deep neural network to minimize the residual stress. Experiments on the drawing process with the conventional die and MRD were performed to evaluate the residual stress and verify the effectiveness of the MRD. The results of X-ray diffraction measurements indicated that the axial and hoop residual stresses on the surface were dramatically reduced.

## 1. Introduction

The drawing process is a processing method that reduces a cross-section by passing materials such as wires, bars, and pipes through one or several dies. Cold-drawn products are actively used in a wide range of industries such as automobile, shipbuilding, aerospace, electric and electronics, off-shore plants, construction, and medical care [1] because of their high strength and good surface quality. During the drawing process, plastic deformation is concentrated on the surface of the material and causes uneven strain and heat generation on the surface and at the center of the material. Therefore, residual stress deviation of the material occurs after drawing [2,3].

The tensile residual stress adversely affects the stress corrosion cracking and fatigue life of the product. Although compressive residual stress enhances the fatigue life and the resistance to stress corrosion cracking, it causes additional damage when the product is subjected to cutting, grinding, or heat treatment [4,5]. Compression residual stress can inhibit the surface crack generation and wear damage, but tensile residual stress has a serious effect on the fatigue strength and fracture of the material [6]. In contrary with the drawing process, rolling processes delay the diffusion of hydrogen towards the inner areas by introducing high compressive residual stresses [2,7]. The absence of residual stresses makes the crack propagate towards a preferential crack path. A residual stress profile with tensions in the vicinity of the wire surface and compressions in the central area makes the crack propagate towards a quasi-straight crack front. Such a profile seems to affect crack propagation more markedly in tension [8]. Consequently, it is important to control the residual stress of the drawn material to improve the quality of the product.

Several researchers have conducted studies on the analysis and measurement of the residual stress of drawn materials. Toribio et al. [9] analyzed the residual stress of a wire surface in which fatigue loading was added in the cold drawing process. He confirmed that redistribution of residual stress in the material occurred owing to fatigue loading. Oh et al. [10] studied the effect of heat treatment conditions and cooling methods on the residual stress of a wire rod after the drawing process.

Lee et al. [11] evaluated the effects of flow stress, reduction in area (R.A), and half die angle on the residual stress in a multi-pass drawing process. Furthermore, they proposed an equation for predicting the surface residual stress according to the R.A and half die angle. Överstam [12] evaluated the effect of residual stress based on the contact type of the die-bearing part in the drawing process. Coser et al. [13] reduced the residual stress by modifying the shape of the die-bearing part. Kuboki et al. [14] presented a new die design, consisting of two consecutive straight tapered portions, to reduce residual stresses in the bar after the drawing process. The results of their study indicated that the double-tapered die could remarkably reduce the axial residual stress. Ko et al. [15] evaluated the effects of the half die angle, R.A, friction factor, and bearing length on the residual stress distribution, which are the main variables of the drawing process.

Kuboki et al. [16] investigated the influence of mechanical properties on the intensity of axial residual stress, measured using the Sachs method, after cold drawing. He found that the distribution of axial residual stress is most sensitive to the work-hardening ratio. Furthermore, a skin-pass drawing condition was presented to reduce the axial residual stress. He et al. [17] predicted the residual stress distribution of an anisotropic material in the drawing process using finite element (FE) analysis and measured the residual stress using X-ray diffraction (XRD) equipment for comparison. Katagiri et al. [18] measured the hardness and residual stress components to compare the effects of bluing and shot peening on the fatigue strength of cold-drawn wires at a wide range of temperatures.

For the conventional techniques presented, additional processes and optimization of process variables are required to reduce the residual stress of the drawn material. Recently, machine learning (ML) has been applied to the optimization of processes. Optimization through ML results in improved product quality in terms of the cost, time, consumption of resources, and specific optimization [19]. Hart-Rawung et al. [20] proposed a prediction model for the phase fraction in a hot stamping simulation using ML to reduce the total calculation time. Stanke et al. [21] used ML to optimize the height of the die roll in the fine-blanking process. Stendal et al. [22] studied the use of neural networks and hybrid approaches to predict the flow stress in steel and other materials. However, the development of a hybrid model that combines ML with a traditional phenomenological model is absent in the literature. Abbassi et al. [23] used ML to obtain a reliable combination of tube formation parameters for producing a T-shaped tube at an optimized cost and design time.

In this study, a multiple reduction die (MRD) was developed to reduce the residual stress deviation in the drawing process. FE analysis was performed based on the reduction ratio of each MRD tip. To verify the effect of the MRD, the residual stresses for the conventional die and MRD were compared. Experiments on the drawing process with the conventional die and MRD were performed to evaluate the residual stress. The MRD variables were optimized using a neural network.

## 2. Drawing Process with the MRD

The MRD set consists of several die tips, die cases, and lubricating equipment, as shown in Figure 1. All the die tips of the MRD were disposed of simultaneously. The R.A in the drawing process gradually decreased with the proposed MRD, resulting in a decrease in the residual stress deviation. Lubrication was continuously performed between the die tips using the lubricating equipment during the drawing process.

### 2.1. Mechanism of the MRD

The compressive residual stress at the surface of the drawn material can be produced through small reductions in the drawing process. Therefore, in the MRD, the R.As of each die tip were controlled.

Figure 2 shows the axial stress at the center and on the surface during the drawing process with a single die and MRD. After the drawing process, the stress components are elastically recovered to balance the stress at the center and on the surface. Consequently, the residual stress was determined. The compressive stress originated at the center of the drawn material, whereas the tensile stress occurred on the surface. The axial stresses increase at the center and decrease on the surface in the deformation zone. In the drawing process with the MRD, the residual stresses gradually reduced as the material passed through the die tips. Finally, the residual stress after drawing was reduced. Thus, the deviation of the residual stress between the center and the surface could be reduced.

### 2.2. FE Simulation

To evaluate the effect of the MRD on the residual stress, a FE simulation was conducted using commercial S/W, DEFORM-2D [24] with an elasto-plastic FE model, as shown in Figure 3. The initial mesh structure of the FE model was constructed with approximately 70,000 initial 4-node quadrilateral elements, and an automatic remeshing scheme was used for the numerical simulations. The dies were considered to be rigid, and the workpieces were pulled in the drawing direction using the sticking boundary condition between the grip and the workpiece. The sticking boundary condition prevents sliding or separation. An isotropic hardening was considered in FE simulations. The mechanical properties of austenitic stainless steel 304L, presented in Table 1, were investigated. Table 1 lists the general conditions for the FE simulation with a single die and MRD.

The residual stresses of the drawn material for the single die and MRD were measured and compared. Double and triple die tips were applied in the MRD for the FE simulation of the drawing process. The total R.A for a single die was approximately 30.56%. The R.A of each die tip in the MRD can be obtained from the total R.A through division. Therefore, the drawn material through the MRD was gradually deformed.

The R.A, which is the main variable in the drawing process, can be expressed using the cross-sectional area of the initial and drawn materials, as follows:(1)$$R.A=\frac{\left(A_{o}-A_{f}\right)}{A_{o}}\times 100\left(\%\right)$$
where *A*_0_ and *A_f_* are the cross-sectional areas of the initial and drawn materials, respectively.

The R.As of the double die tip were defined as shown in Table 2. The axial and hoop residual stresses were compared, as shown in Figure 4. When the 1st die tip had a low R.A, the axial residual stress on the surface of the material was shifted towards tension. By contrast, when the 1st die tip had a high R.A, the axial residual stress on the surface was shifted towards compression, and the deviation of the axial residual stress between the center and surface of the drawn material decreased. The hoop residual stress at the center was shifted towards tension for all cases, as shown in Table 2. When the 1st die tip had a low R.A, the hoop residual stress on the surface was shifted towards tension by the MRD. When the 2nd die tip had a high R.A, the hoop residual stress on the surface was shifted towards compression and the hoop residual stress at the center was shifted towards compression. Therefore, the results of the FE simulation reveal that compressive residual stress can be produced by a slight reduction in the final die tip in the MRD.

The R.As of the triple die tip were defined as shown in Table 3. Figure 5 presents the axial and hoop residual stresses of the drawn material by the triple die tip. The results of the FE simulation indicated that the R.A of the last die tip was dominant in controlling the residual stress in the drawing process with the MRD. Similar to the drawing process with the double die tip, the axial and hoop residual stresses were reduced by a small reduction in the 3rd die tip. When the 3rd die tip had a smaller R.A, the axial residual stress was shifted towards tension at the center and shifted towards compression on the surface of the drawn material. Therefore, a considerable reduction in the 1st die tip is required to minimize the R.A of the last die.

Figure 6 presents the FE results for the drawing process with a single die and MRD. The residual stresses of the drawn material were measured as shown in Figure 4 and Figure 5. The compressive and tensile residual stresses originated at the center and on the surface of the drawn material, respectively. Compared with the residual stresses of the single die and MRD, the axial and hoop residual stresses could be reduced by controlling the R.A of each die tip. It was found that the residual stress was dramatically decreased in the drawing process with a triple die tip. A minimum R.A of the last die tip can be ensured for the MRD with the triple die tip.

Several drawing processes have been conventionally performed, and the final drawing step with a very small R.A (termed the “skin pass”) has been performed to control the residual stress distribution. The MRD presented in this study leads to the effect of the skin pass in the drawing of a single pass.

## 3. Experimental Results

### 3.1. Drawing Process with MRD

Experiments on drawing processes with a single die and MRD were performed. As shown in Figure 7, the triple die tips of the MRD were set in tandem for the drawing process. The total R.A of both processes was approximately 30.56%. The applied material was stainless steel 304L, and the initial diameter and length were 12.0 and 1000 mm, respectively. The drawing speed was 3 m/min. A pointing process was performed on the initial material end, and full annealing of the material was performed as a heat treatment before the drawing process. Lubricant (Hangstefer’s Aldraw J-2 dark, Richmond, VA, USA) was supplied between each die tip of the MRD using the lubricating equipment.

### 3.2. Measurement of Residual Stress

Methods for quantitatively measuring the residual stress are largely classified into destructive and nondestructive methods. The residual stress on the surface of sample during the measurement can be changed by the destructive testing methods. Therefore, the destructive testing methods are inappropriate to evaluate the residual stress of the drawn material. Nondestructive testing methods have been developed to prevent damage to the samples [25]. Nano-indentation test has been widely used for measurement of local hardness [26,27]. The axial residual stress is quantitatively evaluated using obtained load-depth curve in the nano-indentation test.

Nondestructive methods mainly measure the residual stress based on the physical properties of the material using the diffraction of the lattice in the material. In this study, XRD was used to measure the residual stress of the drawn material, as it is the most widely used method and has been applied in various measurement fields. XRD is based on change in the interplanar spacing owing to the residual stress in the sample. Thus, the diffraction peak moves when Bragg diffraction occurs, and the magnitude of the movement is related to the stress.

The residual stress was calculated using the following equations:(2)nλ=2dsinθ
(3)σ=KM
where *n* is the order of diffraction, *λ* is the X-ray wavelength, *d* is the plane spacing of the crystal structure, and *θ* is the XRD angle. *K* is the stress constant, and *M* is the slope of the diffraction angle to sin^2^*θ*.

The specimens for measuring the residual stress after the drawing experiments were prepared. While preparing the specimens, it is very important to minimize the deformation of the material caused by cutting the drawn material. Any deformation of the material after drawing causes a change in the residual stress. The length of each specimen was approximately 280 mm. The effectiveness of the FE analysis was verified by quantitatively measuring the residual stress on the surface.

XRD equipment (Stresstech’s Xstress-3000 + G3, Vaajakoski, Finland) was used to measure the residual stress, as shown in Figure 8. Depending on the measurement direction, 0° was considered the axial direction, and 90° the tangential direction. The measurement conditions are listed in Table 4.

The measurement results are presented in Table 5. The axial and hoop residual stresses on the surface of the drawn material were approximately 538.7 and 461.6 MPa, respectively, while using the single die tip and approximately 182.3 and 11.8 MPa, respectively while using the triple die tip. This indicates that the residual stresses were dramatically reduced. The comparison of the measured and predicted stresses demonstrates the effectiveness of the FE analysis.

## 4. Discussion

### 4.1. Optimization of the MRD

In this study, an artificial neural network (ANN) of a multi-layered perceptron type was used to predict the residual stress according to the MRD variables.

The feed forward back propagation, as shown in Figure 9a, with a rectified linear unit (ReLU) active function, was used for the hidden layer of a deep neural network (DNN). The neural network is processed as follows [20]:(4)$$Y_{j}=f\left(\overset{n}{\underset{i=1}{\sum }}Y_{i}w_{ij}+b_{j}\right)$$
where *Y_j_* is the output of the node, *f* is the activation function, *n* is the number of nodes, *Y_i_* is the output of the previous layer node, *w_ij_* is the weight, and *b_j_* is the bias of the node.

In the learning process, the network is presented with an input pattern and a corresponding desired output pattern. Using the weights and thresholds, the network produces an output pattern that is compared with the desired output pattern. The active function is represented as follows [28]:(5)$$f\left(x\right)=max\left(0,x\right)$$
where *f* is the activation function, and *x* is the input data.

The data used to train a neural network are commonly divided into three subsets: a training set, a testing set, and a validation set. The training set was used to adjust and optimize the ANN parameters (i.e., weights and biases) during the training. The validation set was used to prevent overfitting. The criterion for early stopping was an increase in the validation error beyond a set threshold with an increasing number of epochs.

In this study, a neural network with 50 hidden layers was employed. The input dataset was divided into three groups: 70% training, 15% testing, and 15% validation. The range of learning is shown in Table 6. The first die tip reduction, second die tip reduction, and half die angle were trained as input data. The residual stress index was learned as the output data. To train the input dataset, a neural network with eight hidden layers was employed in this study, and 120 samples were used. The Latin hypercube sampling method as shown in Figure 9b, which can obtain high accuracy with a small number of samples, was used [25].

The index of residual stress was introduced to evaluate the distribution of the axial residual stress quantitatively:(6)$$\mathrm{Index}=\int_{0}^{d/2}2\pi \left|\sigma_{z}\right|rdr$$
where *d* is the diameter of the drawn material, *σ_z_* is the axial residual stress, and *r* is the distance from the center. A low index indicates a low residual stress deviation between the center and surface and low tensile residual stress on the surface. Therefore, the DNN was performed to minimize the index of residual stress.

The performance was evaluated using the coefficient of determination (*R*^2^) and mean absolute error (MAE). *R*^2^ is given by [29]
(7)$$R^{2}=1-{\left(\frac{\sum \left(y-\bar{y}\right)}{\sum \left(y-\hat{y}\right)}\right)}^{2}$$
where y is the input data, $\hat{y}$ is the predicted data from the DNN, and $\bar{y}$ is the average value of the predicted data.

The MAE value can be determined as [29]
(8)$$MAE=\frac{1}{n}\sum \lceil \hat{y}-y\rceil$$
where *n* is the number of samples, y is the FE result, and,$\;\hat{y}$ is the predicted value from the DNN.

The MAE was utilized as the criterion for early stopping in this study, as shown in Figure 10. The minimum MAE was obtained at an epoch of 126. The performance of the neural network was evaluated using *R*^2^. The Adam optimization algorithm was used to reduce errors by optimizing the weights for its simple implementation and high computational efficiency [30]. As shown in Figure 11, the *R*^2^ values were 0.9792, 0.9858, 0.9842, and 0.9802 for the training, validation, testing, and all datasets, respectively.

As a result of the DNN, the index of the residual stress according to the reduction in each die tip was predicted, as shown in Figure 12a. The minimum index of residual stress according to the half die angle is shown in Figure 12b. The R.A of each die tip for minimizing the index of the residual stress was predicted by the DNN. It was shown that the index of residual stress can be decreased by decreasing half die angle for the same total strain in drawing process. The R.A of the 1st die tip increased and that of the last die tip decreased, and the index of the residual stress decreased; the R.As of the 1st, 2nd, and 3rd die tips for the half die angle of 10° were 21.38, 7.5, and 1.68%, respectively. It was shown that artificial neural networks can be used to design the MRD with very high accuracy and that good results can be obtained with smaller data sets. In future work it will be studied how experimental data can be used to reduce the remaining error between FE analysis and experiment.

### 4.2. Comparison between the Single Die and MRD

FE analysis was performed for the drawing processes with a single die and the optimized MRD. A half die angle of 10° was applied to a single die. As shown in Figure 13, the axial stresses of the center and surface during the drawing process were measured to compare the single die and MRD.

Comparing the single die and the triple die, as shown in Figure 13a, the compressive and tensile stresses in the axial direction were reduced at the die exit. The axial stress was increased at the center and was decreased on the surface in the deformation zone. In the drawing process with the triple die, the residual stresses were gradually reduced as the material passed through the die tips. Finally, the axial and hoop residual stresses were dramatically reduced in the drawn material owing to the MRD, as shown in Figure 13b. In addition, the distribution of residual stress was flattened. Experiments were performed on the drawing process. The residual stress was measured using XRD. The drawn materials obtained by the drawing process of the single die and the optimized MRD were compared, as shown in Table 7. The axial residual stresses were reduced from 538.7 ± 28.5 MPa to −41.2 ± 20.6 MPa, whereas the hoop residual stresses were reduced from 461.6 ± 55.1 MPa to −203.1 ± 50.7 MPa owing to the MRD. The compared results indicate that the axial and hoop residual stresses on the surface can both be reduced.

## 5. Conclusions

In this study, an MRD was developed to improve the distribution of residual stresses. From the FE results, the distribution of the residual stresses was summarized as an index. Moreover, the MRD was optimized using a DNN. The following conclusions were drawn:(1)Under the same drawing conditions, the single die and MRD were compared using FE analysis. The double die tip exhibited a maximum residual stress index of 41.82%. For the triple die tip, the index of the residual stress was reduced to a maximum of 77.41%. The index of axial residual stress was reduced by an average of 47.59% through a small reduction in last die of triple die.(2)The experimental results of the drawing process with the optimized MRD revealed that the axial and hoop residual stresses on the surface were dramatically reduced compared to the residual stresses for a single die.(3)Drawing processes were performed with a single die and the optimized MRD. The indexes of the axial and hoop residual stresses on the surface of the drawn material were reduced by a maximum of 94.82 and 69.89%, respectively.(4)Axial stresses during the drawing process with a single die and the optimized MRD were compared. The stress during the process was reduced by the MRD. Finally, it was demonstrated that the residual stress could be reduced by the MRD.

## Figures and Tables

**Figure 1 materials-14-01358-f001:**
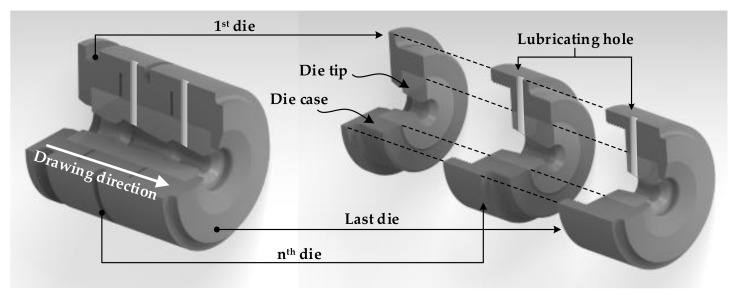
Schematic illustration of the multiple reduction die (MRD).

**Figure 2 materials-14-01358-f002:**
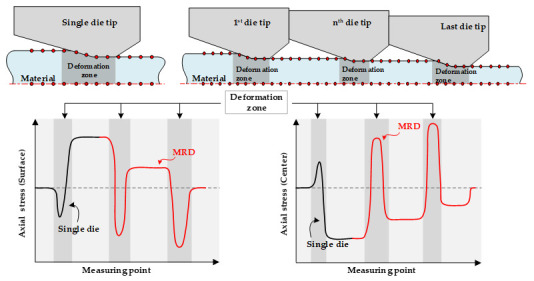
Schematic illustration of axial stress in the drawing process with MRD.

**Figure 3 materials-14-01358-f003:**
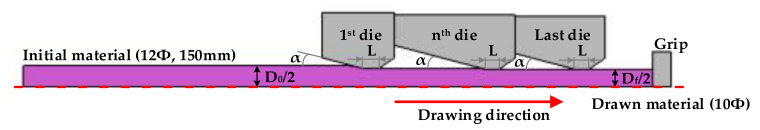
Finite Element (FE) model for drawing process with MRD.

**Figure 4 materials-14-01358-f004:**
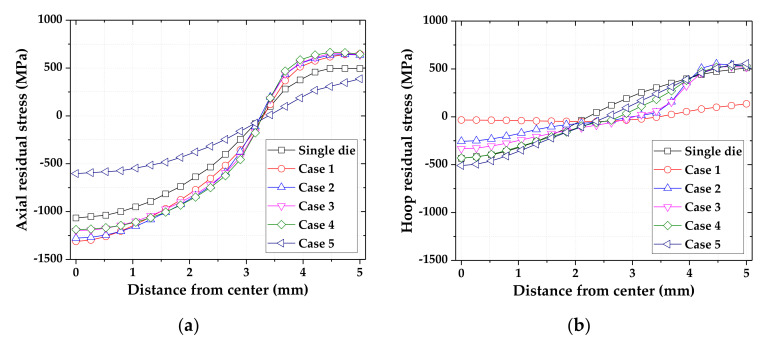
Residual stress of drawn material by double die tips according to the reduction in area (R.A): (**a**) axial residual stress; (**b**) hoop residual stress.

**Figure 5 materials-14-01358-f005:**
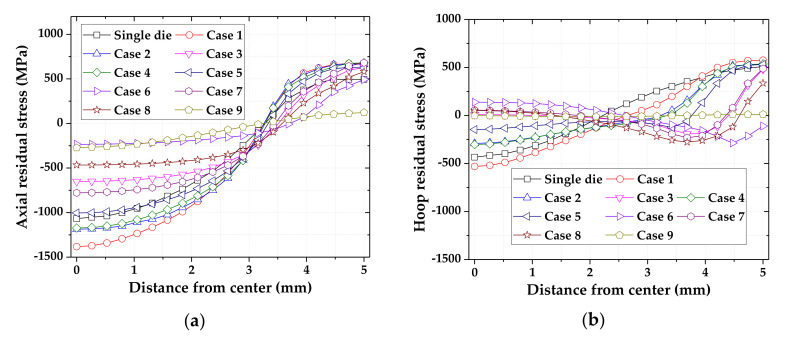
Residual stresses of drawn material by triple die tips according to R.A: (**a**) axial residual stress; (**b**) hoop residual stress.

**Figure 6 materials-14-01358-f006:**
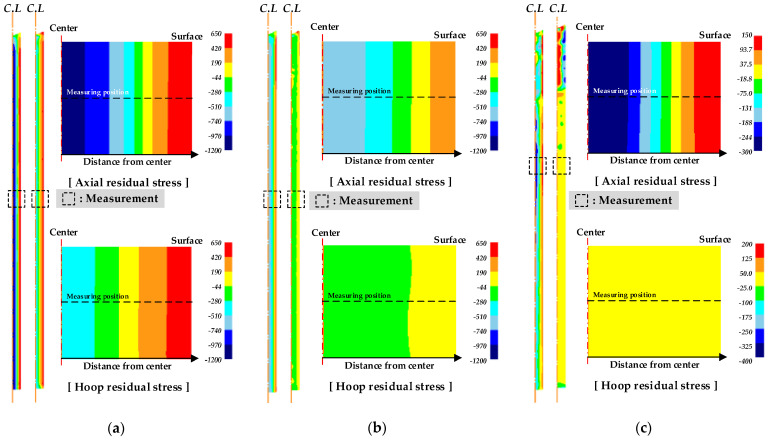
Measurement of axial and hoop residual stresses from the FE results for drawing process: (**a**) single, (**b**) double (Case 5), and (**c**) triple dies (Case 9).

**Figure 7 materials-14-01358-f007:**
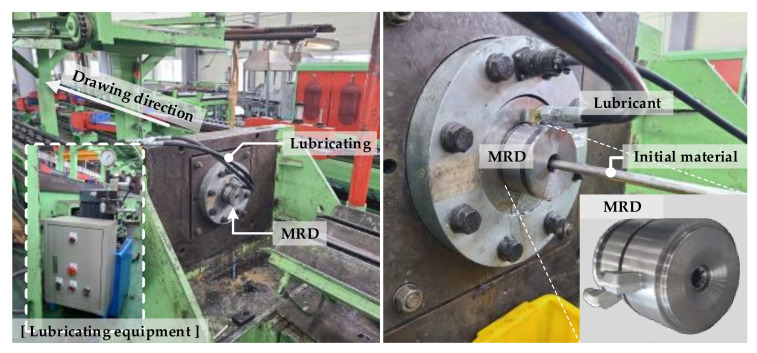
Experiments on the drawing process.

**Figure 8 materials-14-01358-f008:**
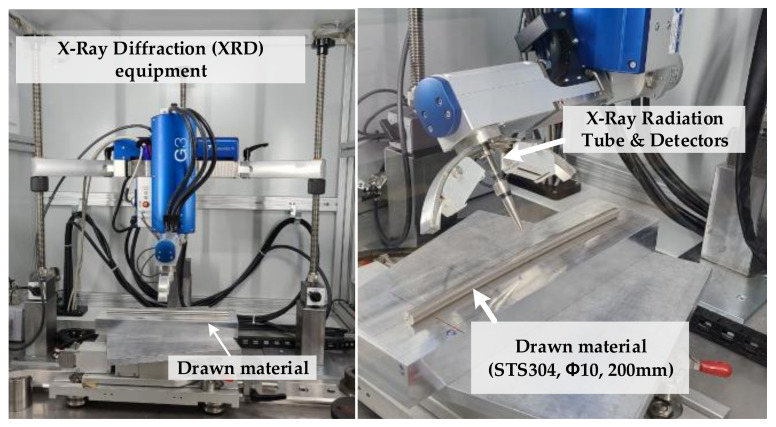
XRD equipment for measuring the residual stress (Xstress-3000 + G3).

**Figure 9 materials-14-01358-f009:**
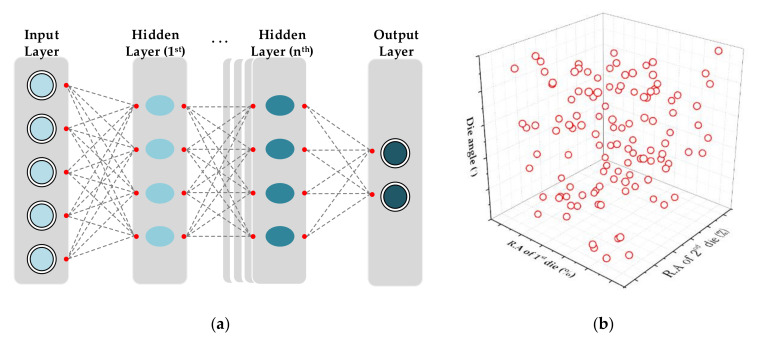
Schematic diagram of sampling method and neural network: (**a**) Deep Neural Network (DNN); (**b**) Latin Hypercube Sampling (LHS).

**Figure 10 materials-14-01358-f010:**
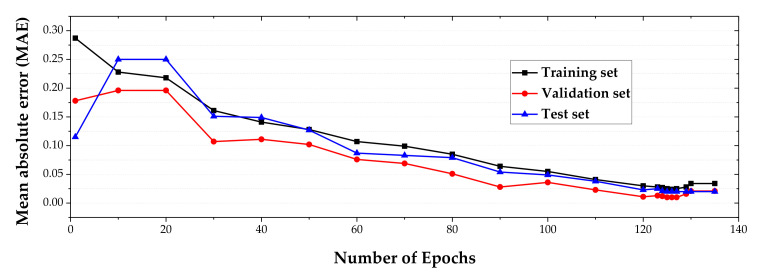
Accuracy evolution of the training, validation, and test and all datasets based on the MAE.

**Figure 11 materials-14-01358-f011:**
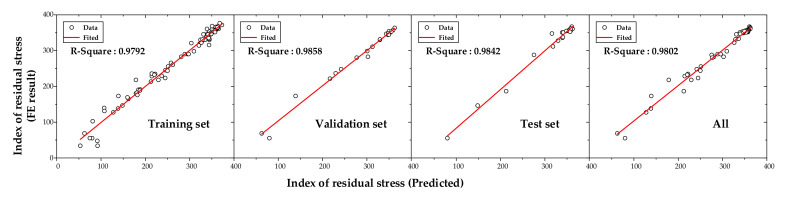
Comparison of the index of residual stress between the FE result and the DNN.

**Figure 12 materials-14-01358-f012:**
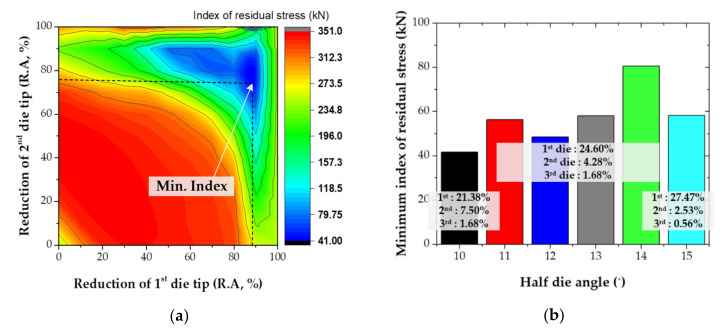
Result of machine learning (ML) according to the half die angle: (**a**) predicted result by DNN; (**b**) minimum indexes.

**Figure 13 materials-14-01358-f013:**
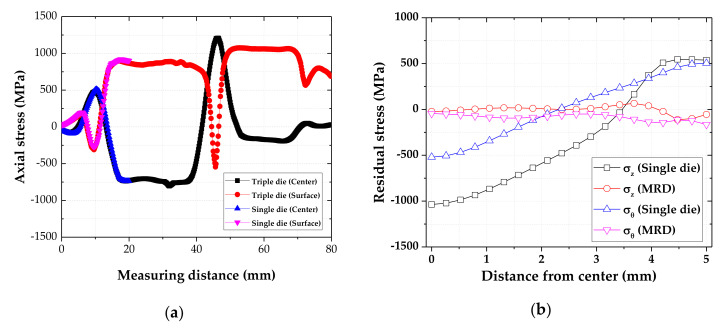
Comparison of stresses of the material drawn using the optimal MRD and the single die: (**a**) axial stress during the drawing process; (**b**) residual stress of drawn materials.

**Table 1 materials-14-01358-t001:** Input parameters required for the drawing process.

Parameters	Value
Material	STS 304L
Flow stress of initial material (MPa)	$\;\bar{\sigma }=1308.6{\;\bar{\varepsilon }}^{0.449}$
Young’s modulus (GPa)	193
Poisson’s ratio (ν)	0.29
Initial diameter (mm, D_0_)	12.0
Final diameter (mm, D_f_)	10.0
Total reduction in area (%)	30.56
Bearing length of die (mm, L)	0.2D_0_
Half die angle (α, °)	15
Friction coefficient (μ)	0.06

**Table 2 materials-14-01358-t002:** R.A of the MRD for the double die tip.

Case No.		1	2	3	4	5
R.A (%)	Total	30.56 (100%)				
	1st tip	3.06 (10%)	9.17 (30%)	15.28 (50%)	21.39 (70%)	27.50 (90%)
	2nd tip	27.50 (90%)	21.39 (70%)	15.28 (50%)	9.17 (30%)	3.06 (10%)

**Table 3 materials-14-01358-t003:** R.A of the MRD for the triple die tip.

Case No.		1	2	3	4	5	6	7	8	9
R.A (%)	Total	30.56 (100%)								
	1st tip	3.056 (10%)	3.056 (10%)	3.056 (10%)	15.28 (50%)	15.28 (50%)	15.28 (50%)	27.50 (90%)	27.50 (90%)	27.50 (90%)
	2nd tip	2.75 (9%)	13.75 (45%)	24.75 (81%)	1.52 (5%)	7.64 (25%)	13.75 (45%)	0.31 (1%)	1.53 (5%)	2.75 (9%)
	3rd tip	24.75 (81%)	13.75 (45%)	2.75 (9%)	13.75 (45%)	7.64 (25%)	1.53 (5%)	2.75 (9%)	1.53 (5%)	0.31 (1%)

**Table 4 materials-14-01358-t004:** Measurement conditions for the XRD equipment.

Conditions	Value
Characteristic X-ray	CrKα
Voltage (kV)	30
Current (mA)	9
Exposure time (s)	40
Collimator (Ø)	3
Diffractive angle (°)	148.9
X-ray wavelength (Å)	2.29107

**Table 5 materials-14-01358-t005:** Comparison of the measured and predicted residual stresses in the single die and MRD processes.

Single Die and MRD		XRD (MPa)	FE Result (MPa)
Single die tip	Axial residual stress	538.7 ± 28.5	531.21
	Hoop residual stress	461.6 ± 55.1	499.81
Triple die tip (Case 9)	Axial residual stress	182.3 ± 42.9	174.7
	Hoop residual stress	11.8 ± 24.8	13.1

**Table 6 materials-14-01358-t006:** Range of learning for the MRD.

Variables	Minimum	Maximum
1st die tip reduction (R.A_1_)	0% of R.A_total_	100% of R.A_total_
2nd die tip reduction (R.A_2_)	0% of (R.A_total_−R.A_1_)	100% of (R.A_total_−R.A_1_)
Half die angle	10°	15°

**Table 7 materials-14-01358-t007:** Comparison of the measured and predicted residual stresses in single die and optimal MRD processes.

Single Die and MRD		XRD (MPa)	FE Result (MPa)
Single die tip	Axial residual stress	538.7 ± 28.5	531.21
	Hoop residual stress	461.6 ± 55.1	499.81
Triple die tip	Axial residual stress	−41.2 ± 20.6	−54.54
	Hoop residual stress	−203.1 ± 50.7	−164.07

## Data Availability

Data sharing is not applicable to this article.

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
