# Peer review of "Effect of a Multiple Reduction Die on the Residual Stress of Drawn Materials"

_materials, 2021, doi:10.3390/ma14061358_

Round 1

Reviewer 1 Report

The manuscript presents a very extensive scientific work. The FEM method, the ANN method and laboratory tests are a very comprehensive solution to the problem of residual stresses in drawn wires. 

The research presented in the manuscript has high scientific value (especially in computer modeling). They will not have high application value in the metal industry. Currently, the reduction of residual stresses is achieved by the skinn - pass drawing method. It is an effective and cheap method. It can be used on any drawing line. In the final conclusions, the manuscript did not provide information on the average numbers of stresses for one, two and three dies. The percentage is also OK, but the average numbers provide more information. Overall, the manuscript is very good.

Reviewer 2 Report

The reviewer’s comments for the paper « Effect of a Multiple Reduction Die on the Residual Stress of Drawn Materials»

- Reviewer

The authors present an article « Effect of a Multiple Reduction Die on the Residual Stress of Drawn Materials». However, there are several points in the article that require further explanation.

Comment 1:

Give the FE model showing geometry dimensions of the billet and die tooling.

Comment 2:

‘The effectiveness of the FE analysis was verified by quantitatively measuring the residual stress on the surface.’

Have the authors validate the FE results by comparing the drawing force? How is the friction coefficient determined in FE? It could have a very large influence on the surface deformation and the drawing force, it would be very important that the friction coefficient is calibrated by comparing the simulated force with the experimental one.

Comment 3:

Besides the reduced residual stresses, are there any other advantages compared with the conventional drawing die? What about the surface quality and mechanical strength?

Comment 4:

In Section 3.2, please very briefly introduce other non-destructive methods, such as nanoindentation, before introducing XRD and justifying the choice of XRD in this paper. Please kindly cite a relevant reference: doi: 10.1016/j.msea.2014.11.018

Comment 5:

Strictly speaking, the ‘Discussion’ section is not discussion. It is more like a combination of methods and results, with very slight discussion. A large part of this section is introduction of methods and description of results. There is very little discussion of the results.

The topic of the article is interesting and deserves attention. It can be seen that the authors have done significant research on the stated topic. However, authors should carefully study all comments. Only after major changes can the article be considered for publication in Materials.

Reviewer 3 Report

This is a well-written paper with a very interesting aim: to obtain the optimum distribution of “reduction areas” in a multiple reduction die process. The combination of FE simulations (even though they are very simple simulations and the FEA insight and novelty are low), XRD measurements (that successfully match FEA predictions) and robust Machine Learning techniques results in a sound research, that should be appealing for both researchers and technicians, with strong implications for manufacturers.

However, as already mentioned, the novelty lies in the combination of methods and not in any specific contribution. Additionally, the physical meaning of the residual stress index (a integration of axial stress in kN) should be better justified. In my opinion, this is the weakest point of the paper. It is not clear what does the minimization of this Index imply and whether this is the optimum for avoiding crack nucleation and propagation. A brief discussion on crack mechanisms, and possibly Fracture Mechanics, could be required.

Therefore, I recommend the publication of the present paper after minor revision. Additionally, the following concerns should be addressed:

  • The following sentence in the first page should be better described, with a longer discussion and some references, since there lies the motivation of the paper: “Cracks can be generated and propagated through axial residual stress during twisting of the drawn material. In addition, cracks are generated and propagated on the surface of the drawn material owing to hoop residual stress.”
  • References cited in text should be uniform: sometimes the first author is only mentioned, e.g. Toribio [6], even though there are more than one author, whereas the note “et al.” is included for other references.
  • The reference “Kuboki [13]” in the text is not correct. All references must be checked.
  • Optimization is also connected to an output variable, and this should be clear since the introduction. Which is the minimised output? The tensile residual stress peak in axial direction? In this case, it should be discussed whether a higher tensile peak is more harmful than a residual stress distribution with a lower peak but higher surface penetration.
  • What do the authors mean by “the strains of the surface and center were adjusted”?
  • FE modelling details should be given. It is not clear the geometrical features of the model (2D, 3D, axisymmetric…). Though the code “DEFORM-2D” is used, it is stated that tetrahedral elements are used. More details on “sticking” contact should be given.
  • I assume that isotropic hardening is considered, but this should be mentioned.
  • It is not clarified whether the numerical scheme is implicit or explicit, which is very relevant.
  • The discussion of Figures 3 and 4 could lead to confusion using the terms “decreasing” or “increasing” residual stresses; it is preferable to express a “shift towards tension or compression”.
  • The following sentence is controversial: 189-190: “The destructive testing methods are relatively simple to execute, and the test accuracy is generally high, but the surface damage is unacceptable.”. In my opinion, the test accuracy is not generally high; it depends on the method. Slitting or sectioning method have many particularities, and the problem is not surface damage, but the precision of cutting and the accurate measurement of deformations. If the authors also include here “semi-destructive” methods, e.g. Hole Drilling, Ring Core …, the statement is even more controversial.
  • What does the author mean by “physical” in the sentence “Nondestructive methods, such as physical testing methods”? All methods use “physical” properties of materials in one way or another. In my opinion, this affirmation is not rigorous.
  • I assume that “Diffractive plate” 311 refers to the crystallographic plane, so it sounds strange to me the word “plate”.

Round 2

Reviewer 2 Report

The authors have improved their manuscript according to the comments. I have no further comments.